# Visualization of Murine Vascular Remodeling and Blood Flow Dynamics by Ultra-High-Frequency Ultrasound Imaging

**DOI:** 10.3390/ijms232113298

**Published:** 2022-10-31

**Authors:** Vincent Q. Sier, Alwin de Jong, Paul H. A. Quax, Margreet R. de Vries

**Affiliations:** 1Department of Surgery, Leiden University Medical Center, 2333 ZA Leiden, The Netherlands; 2Einthoven Laboratory for Experimental Vascular Medicine, Leiden University Medical Center, 2333 ZA Leiden, The Netherlands

**Keywords:** ultrasound imaging, vein graft failure, arteriovenous fistula failure, vascular remodeling, animal model, in vivo imaging, vascular access

## Abstract

Vein grafts (VGs) are used to bypass atherosclerotic obstructions and arteriovenous fistulas (AVFs) as vascular access for hemodialysis. Vascular remodeling governs post-interventional arterialization, but may also induce VG and AVF failure. Although the endpoint characteristics of vascular remodeling are known, the in vivo process and the role of blood flow dynamics has not been fully studied. Therefore, here we non-invasively quantify vascular remodeling and blood flow alterations over time in murine VG and AVF models. C57BL/6J (*n* = 7, chow diet) and atherosclerosis-prone ApoE3*Leiden (*n* = 7) mice underwent VG surgery. Ultrasound imaging was performed at 3, 7, 14, 21, and 28 days post-surgery. C57BL/6J mice (*n* = 8) received AVF surgery. Ultrasound imaging was performed at 7 and 14 days post-surgery. The luminal volume increased by 42% in the VGs of C57BL/6J and 38% in the VGs of ApoE3*Leiden mice at 28 days relative to 3 days post-surgery. Longitudinally, an 82% increase in wall volume and 76% increase in outward remodeling was found in the ApoE3*Leiden mice, with a constant wall size in C57BL/6J mice. Proximally, the pulsatility index, resistive index, and peak systolic velocity decreased longitudinally in both groups. Distally, the maximum acceleration increased with 56% in C57BL/6J VGs. Among the AVFs, 50% showed maturation after 7 days, based on a novel flow-criterium of 23 mL/min. Distinct flow patterns were observed at the anastomotic site and inflow artery of the AVFs relative to the control carotid arteries. Vascular remodeling can be quantified by ultra-high-frequency ultrasound imaging over time in complex animal models, via three-dimensional structural parameters and site-specific hemodynamic indices.

## 1. Introduction

Vascular remodeling represents a wide range of vascular wall adaptations to altered hemodynamics and is a crucial mechanism that relates to cardiovascular disease outcomes [1,2]. In particular, vascular remodeling plays a predominant role in post-interventional arterialization in venous bypass grafts and arteriovenous fistula (AVF) maturation [1,3]. Vein grafting is a solution to bypass arterial obstructions caused by atherosclerosis [4]. Directly after vein engraftment, wall distention and increased hemodynamic forces initiate vascular remodeling and the subsequent thickening of the blood vessel. Initially, this thickening of the vessel wall is a beneficial necessary process. In later stages, arterialization of the conduit can eventually develop into either negative inward remodeling or positive outward remodeling [1,5]. While the latter preserves luminal flow capacity, inward remodeling causes lumen loss due to smooth muscle cell-rich intimal hyperplasia or (accelerated) atherosclerosis. This severely reduces long-term patency rates, and leads to vein graft failure [4]. Vein grafted C57BL/6J and atherosclerosis-prone ApoE3*Leiden mice on a high-fat/cholesterol diet are suited to study the processes of intimal hyperplasia and accelerated atherosclerosis, respectively: two important mechanisms in vein graft remodeling. More specifically, the ApoE3*Leiden mice vein graft model involves more profound wall thickening and foam cell accumulation, as shown by post-mortem (immuno-)histological analyses [6]. In AVFs, which serve as access sites for hemodialysis needed in patients with chronic kidney diseases, an early arterialization response is crucial to achieve maturation [7]. Both impaired outward remodeling and excessive intimal hyperplasia form important causes of AVF failure and can be studied in the C57BL/6J model [8,9,10].

The non-invasive nature of ultrasound imaging systems allows for the monitoring of crucial processes in animal disease models, including the identification of blood flow patterns and structural analyses [11]. In particular, ultra-high-frequency ultrasound imaging offers the advantage of higher resolutions (30 μm vs. 300 μm) as compared to conventional ultrasound systems, with adequate frame rates and depth. Combined with pulsed wave (PW) Doppler, which measures the reflection of sound waves on moving objects, ultrasound imaging allows for the quantification of structural as well as hemodynamic parameters, including velocity indices such as the pulsatility and resistive index (PI, RI) [12,13]. In the clinical setting, ultrasound imaging is used to monitor vascular remodeling, as is illustrated by its adoption as a detection and follow-up tool in patients with an aortic aneurysm or peripheral artery disease [14,15,16,17]. Preclinically, ultrasound imaging has been widely employed, ranging from the quantification of angiogenesis in murine tumor models to a two- and three-dimensional vessel wall and lumen reconstructions [18,19,20]. In comparison to typical endpoint histological analyses for vascular remodeling, the ability of ultrasound imaging to follow these processes over time offers both a reduction in animal use and an improved representation of the in vivo setting [20]. The current study examined, using state-of-the-art ultrasound imaging technology, vascular remodeling and blood flow dynamics over time in a complex murine vein graft and AVF models. Novel and clinically relevant parameters were established to quantify murine vein graft and AVF remodeling over time.

## 2. Results

### 2.1. Area and Volume Ultrasound Measurements of Vein Grafts of C57BL/6J and ApoE3*Leiden Mice Can Be Quantified over Time

#### 2.1.1. Two-Dimensional Morphological Analysis

To examine the potential of ultrasound imaging to quantify vascular remodeling over time, vein grafts from C57BL/6J and ApoE3*Leiden were measured by ultrasound (Figure 1A). The murine models, respectively, represent physiologic and pathologic accelerated atherosclerosis in vein graft remodeling [6]. The measurements of total vein graft areas, wall areas, and lumen areas were performed in short-axis B-mode recordings (Figure 1B).

The lumen and wall area of the vein grafts of C57BL/6J and ApoE3*Leiden mice were measured over time. On 28 days post-surgery, an increase of 29% (*p* = 0.003) and 24% (*p* = 0.008) in the lumen area of, respectively, the C57BL/6J and ApoE3*Leiden mice was found relative to the baseline measurement on day 3 (Figure 1C). More specifically, the lumen areas of the vein grafts in C57BL/6J mice were also significantly increased by 23% at 21 days compared to 3 days post-surgery (Figure 1C; *p* = 0.01). The wall area, a parameter of intimal hyperplasia, significantly increased by 35% in the ApoE3*Leiden mice between 3 and 21 days (*p* = 0.0005) and by 46% between 3 and 28 days post-surgery (Figure 1D; *p* < 0.0001). The wall area of the vein grafts in C57BL/6J was also measured and found to be constant over time. Correspondingly, a 56% and 33% significantly larger wall area was observed in the vein grafts in ApoE3*Leiden mice compared to the vein grafts in C57BL/6J mice, respectively, 14 (*p* = 0.01) and 28 (*p* = 0.01) days after surgery (Figure 1D).

Outward remodeling was quantified via the total vein graft area, consisting of the sum of the wall and lumen area. Consistent with an increased wall and lumen area, the total vein graft area in the vein grafts of ApoE3*Leiden mice significantly increased by 43% over the course of 28 days (Figure 1E; *p* < 0.0001). Moreover, a significant 32% increase was found in the total area of the vein grafts of the ApoE3*Leiden mice at 21 days (*p* = 0.0006), while the total vein graft area of the C57BL/6J mice remained stable during the 28 days post-surgery (Figure 1E). Furthermore, the total vein graft area was increased by 47% after 14 days and 29% at 28 days post-surgery in the ApoE3*Leiden mice vein grafts relative to the C57BL/6J group (Figure 1E; *p* = 0.027, *p* = 0.028).

#### 2.1.2. Three-Dimensional Reconstructions for Volume Measurements

To demonstrate the capacity of ultrasound imaging to quantify the extent of vascular remodeling in vein grafts from C57BL/6J and ApoE3*Leiden mice over time, three-dimensional short-axis B-mode and/or color Doppler ultrasound recordings were analyzed (Figure 2A,B). Here, we introduce a novel parameter for vascular remodeling, representing three-dimensional remodeling corrected for horizontal extension.

Both the vein grafts in C57BL/6J and ApoE3*Leiden mice showed a significant increase in relative lumen volume over time, from 100% on day 3 to, respectively, 142% and 138% on day 28 post-surgery (Figure 2C; *p* = 0.01, *p* = 0.005). From 3 to 28 days after surgery, an 85% increase in wall volume and a 76% in total vein graft volume were observed in the vein grafts of the ApoE3*Leiden mice (Figure 2D,E; *p* < 0.0001, *p* < 0.0001). Volume measurements in the vein grafts of the C57Bl/6J mice showed a constant wall and total vein graft volume over the course of 28 days. Accordingly, a significant 2-fold larger wall volume was found on day 28 in the vein grafts in ApoE3*Leiden mice relative to the C57BL/6J group (*p* < 0.0001), which was substantiated by a 90% larger relative total vein graft volume (Figure 2D,E; *p* < 0.0001). Consequently, the wall-to-lumen ratio differed significantly between the vein grafts in both groups at 28 days post-surgery, with a value of 1.4 in the ApoE3*Leiden and 0.7 in the C57BL/6J mice (Appendix A).

Altogether, the two-dimensional area and novel three-dimensional volume ultrasound imaging measurements precisely demonstrate the alterations in structural vascular remodeling over time.

### 2.2. Clinically Relevant Ultrasound Velocity Measurements and Derivatives Can Be Quantified at Three Distinct Sites in Vein Grafted C57BL/6J and ApoE3*Leiden Mice over Time

To follow the contribution of blood flow velocity to vascular remodeling at specific sites of the vein grafts of C57BL/6J and ApoE3*Leiden mice over time, PW Doppler recordings were used to calculate multiple clinically relevant velocity indices. These velocity indices were derived from equations that include the mean, peak, and end-diastolic velocity (Figure 3A). The measurements of blood flow velocity can provide specific information on flow-remodeling interactions [21].

Cumulative (i.e., mean proximal, middle, caudal) PI, RI, and PSV were quantified over time in vein grafts of C57BL/6J and ApoE3*Leiden mice (Appendix A). In the vein grafts of ApoE3*Leiden mice, early decreases in cumulative PI were found between 7 and 14 days and 7 and 21 days post-surgery (Appendix A; *p* = 0.031, *p* = 0.014). Similarly, significant decreases in cumulative RI were observed between 7 and 14 days and 7 and 21 days after surgery in the vein grafts of ApoE3*Leiden mice (Appendix A; *p* = 0.028, *p* = 0.016). Whereas significant differences were identified in cumulative RI and PI over time, cumulative PSV remained stable in the vein grafts of C57Bl/6J and ApoE3*Leiden mice for 28 days (Appendix A).

Upon splitting the cumulative data in proximal, middle, and distal measurements in the vein grafts, it became apparent that time-bound changes primarily originated in the proximal part of the vein grafts (Figure 3B–D, Appendix A). The proximal PI remained constant in the vein grafts of C57BL/6J mice, while a significant decrease of 27% was found in the vein grafts of ApoE3*Leiden mice at 21 days in comparison to 3 days post-surgery (Figure 3B; *p* = 0.043). The proximal RI decreased significantly by 12% in the vein grafts of ApoE3*Leiden mice at 21 days in comparison to 3 days post-surgery (Figure 3C; *p* = 0.048). Regarding the proximal PSV, a decrease was observed over time in the vein grafts of C57BL/6J and ApoE3*Leiden mice, respectively, to 45% from 14 to 28 days and 45% from 7 to 28 days post-surgery (Figure 3D; *p* = 0.001, *p* = 0.003). A 2-fold higher proximal PSV was found in the vein grafts of C57BL/6J mice relative to the ApoE3*Leiden mice at 14 days post-surgery (Figure 3D; *p* < 0.001). The spike in PSV in the proximal part of the C57BL/6J vein grafts at 14 days post-surgery coincided with a significant increase in the relative flow rate at that time point (Appendix A).

Overall, cumulative and site-specific RI, PI, and PSV measures were quantified in the vein grafts of C57BL/6J and ApoE3*Leiden mice over the course of 28 days. Differences over time were primarily identified in the proximal part of the vein grafts. Furthermore, the proximal relative PSV demonstrated distinct flow patterns between the vein grafts of C57BL/6J and ApoE3*Leiden mice.

### 2.3. Systolic Acceleration Measurements in Vein Grafts of C57BL/6J and ApoE3*Leiden Mice over Time

Acceleration measurements are used in the clinic for vascular remodeling [22,23], and have been previously inaccessible in mice. To longitudinally quantify blood flow acceleration in vein grafts of C57BL/6J and ApoE3*Leiden, ACCsys and ACCmax were identified (Figure 4A) [24,25].

The cumulative relative ACCsys of the vein grafts of the C57BL/6J and ApoE3*Leiden mice remained stable for 28 days (Figure 4B, Appendix A). The relative ACCmax showed a different pattern, with an increase of 56% at 28 days compared to 3 days post-surgery in the C57BL/6J group (*p* = 0.035), as opposed to a constant pattern in the vein grafts in ApoE3*Leiden mice over time (Figure 4C). Accordingly, the relative ACCmax of the vein grafts in the C57BL/6J mice was significantly increased by 2-fold at 28 days post-surgery in comparison to the vein grafts in the ApoE3*Leiden mice (Figure 4C; *p* = 0.003).

By measuring at specific sites within the vein grafts between 3 and 28 days post-surgery, a stable ACCmax was observed in the proximal and middle part of the vein grafts of C57BL/6J and ApoE3*Leiden mice (Figure 4D,E). However, a significant increase of 141% was found in the distal relative ACCmax in the vein grafts of C57BL/6J mice between 3 and 28 days post-surgery (Figure 4F; *p* = 0.0027). At 28 days after surgery, the distal relative ACCmax was a significant 2.6-fold larger in the vein grafts of the C57BL/6J mice relative to the ApoE3*Leiden mice.

In conclusion, the ACCsys and ACCmax were quantified at the proximal, middle, and caudal sites of the vein grafts of C57BL/6J and ApoE3*Leiden mice over the course of 28 days using ultrasound imaging. A significant increase in ACCmax was observed in the vein grafts of C57BL/6J mice, which was primarily driven by the distal increase in ACCmax.

### 2.4. Longitudinal Characterization of Flow Patterns in Murine AVFs

#### 2.4.1. Blood Flow Rate Measurements Can Distinguish between Matured and Non-Matured AVFs in C57BL/6J Mice

AVF maturation is crucial to ensure vascular access and is driven by fierce hemodynamic changes. Therefore, the diameter, flow rate, and spectral broadening of AVFs of C57BL/6J mice were determined on days 7 and 14 after surgery and juxtapositioned with data from the common carotid artery (CA) of control C57BL/6J mice (Figure 5A,B). In this model, t = 7 days post-surgery is considered as the time point at which AVF maturation should be reached.

At the site of the AVF anastomosis, the lumen diameter remained constant between 7 and 14 days post-surgery, while a trend towards a 37% decrease in absolute flow was found (Figure 5C,D; *p* = 0.067). Since a 10–20-fold increase in blood flow rate is considered necessary for sufficient AVF maturation in humans [26,27], a cut-off value of ten times the CA flow (10 × 2.3 mL/min = 23 mL/min) was established (Figure 5D). Based on this criterium, two distinct groups consisting of four AVFs were observed, equally distributed as matured and non-matured AVFs at around the cut-off value of 23 mL/min (Figure 5D,E).

#### 2.4.2. Blood Flow Pattern and Velocity Measurements Can Be Used to Quantify the Blood Flow in AVFs in C57BL/6J Mice over Time

To assess the flow disturbance at the site of anastomosis in both blood vessels, an important contributor to AVF maturation, spectral broadening was determined as an index normalized to the PSV. Following the AVFs over time, a stable SBI was observed at the site of the anastomosis as well as the inflow artery at 7 and 14 days post-surgery (Figure 6A,B).

Taking into account the reduction in flow rate in our murine AVF model and the clinical significance of velocity indices for the identification of flow capacity and patterns, the PI, RI, PSV were determined in the inflow artery [23,28,29,30]. An electrocardiogram trigger and respiration gating were applied, and it was demonstrated that the cardiac and respiratory rate did not correlate with the PSV, a major component of the RI and PI (Appendix A). The absolute and relative PSV in the inflow artery could be observed and remained constant over time (Figure 6C, Appendix A). Likewise, the relative RI and PI were obtained at the inflow artery in the AVF of C57BL/6J mice, with a stable pattern between 7 and 14 days after surgery (Figure 6D,E, Appendix A). At the site of anastomosis, similar results were obtained with no differences in absolute and relative PSV, RI, and PI over time (Appendix A). However, when contrasting the data of the AVFs to the control CAs, vastly different flow patterns could be observed (Figure 5B), as shown by the absolute differences in blood flow rate, SBI, PSV, PI, and RI (Figure 5D and Figure 6B–E).

Overall, important clinically relevant blood flow and velocity parameters were obtained in the AVFs of C57BL/6J mice over time. Moreover, AVFs showed a distinguished morphologic and velocity pattern in ultrasonic analyses, and a novel non-maturation/maturation parameter was established with a cut-off value of 23 mL/min.

## 3. Discussion

In this study, we used ultra-high-frequency ultrasound imaging to analyze vascular remodeling in murine vein graft and AVF models over time. Two-dimensional area measurements were performed and novel three-dimensional volume reconstructions quantified, demonstrating distinct vein graft remodeling processes in vein grafts of C57BL/6J and ApoE3*Leiden mice. Furthermore, the blood flow rate was longitudinally quantified in AVF mice and a new cut-off criterium for AVF maturation was established. In the vein graft as well as the AVF models, clinically relevant velocity parameters such as the PI were quantified over time at specific sites of interest.

Ultrasound imaging provides three advantages relative to endpoint histology vascular remodeling quantification. Firstly, ultrasound imaging allows for the longitudinal quantification of vascular remodeling in two- and three dimensions. Our previous research showed that two-dimensional ultrasound vein graft lumen, wall, and total vein graft area measurements were confirmed by histological assessment [20]. In addition, our current two- and three-dimensional data coincide with those from another study in which histologic analyses of vein grafts of ApoE3*Leiden mice showed similar outward remodeling 28 days post-surgery [6]. Secondly, ultrasound imaging enables the follow-up of particular vein grafts in mice over time in their physiological environment, reducing (i) experimental animal use, (ii) inter- and intra-mouse discrepancies, and (iii) external influences such as tissue processing. Thirdly, ultrasound imaging offers additional tools to quantify vascular remodeling in the field of blood flow analyses via its Doppler functionality.

The two-dimensional area and novel three-dimensional volume parameters for vascular remodeling provide new longitudinal insights. By putting structural and hemodynamic ultrasound parameters side by side, previously unavailable interrelationships can be drawn in the context of vascular remodeling. A notable relationship was found between the increased distal ACCmax in the vein grafts in ApoE3*Leiden as compared to C57BL/6J mice and their three-dimensional morphology. On a mechanistic level, the smaller wall-to-lumen ratio in the vein grafts in C57BL/6J as compared to ApoE3*Leiden mice, under conditions of equal lumen size, could explain the increased acceleration rates as observed at the distal end of the vein graft. As described in the literature, thicker vessel walls are related to a reduction in elasticity [31,32]. A larger lumen and smaller wall size in the vein grafts of C57BL/6J mice may reflect increased elasticity, which increases the propulsion of the blood through the vein graft at the distal end. At a cellular level, the accumulation and activity of different cell types in the vessel wall, such as foam cells and smooth muscle cells, are known to alter the vessel’s stiffness [33]. In this context, simultaneous photoacoustic and ultrasound imaging may be used to characterize vascular wall composition, and can be used simultaneously with ultrasound imaging to quantify vein graft disease [34]. Regarding the PSV, a shifted proximal pattern was found between the vein grafts in the C57BL/6J and ApoE3*Leiden mice between 7 and 14 days post-surgery, including an increased PSV in the vein grafts of the C57BL/6J group. This correlates well with the observed surge in flow rate and its encompassing mean velocity in the vein grafts of C57BL/6J mice at the same time point. In humans, a similar effect can be observed in different carotid artery segments, where the remodeling pattern and velocity measurements are segment-specific [35,36,37].

Physiologic alterations during in vivo imaging, such as anesthesia depth and cardiac cycle, may confound ultrasound data [38,39]. In addition to following standardized protocols, ultrasound imaging while simultaneously registering vital parameters such as heart rate, respiratory rate, and electrocardiography allows for the management of possible confounders [40]. To compensate for these confounders in our morphometrical data, an electrocardiogram trigger and respiration gating were applied. Moreover, there was no correlation between heart rate, respiratory rate, and PSV, the latter being a principal component of the used velocity indices (Appendix A).

In the clinic, duplex ultrasound is used to determine the RI, PSV, and blood flow rate in the inflow artery for AVF assessment [41,42]. In our murine AVF model, we were capable to measure these velocity-pertaining ultrasound parameters at both the inflow artery and anastomotic site. Analogous to the human situation, a 10-fold increase in murine blood flow rate (23 mL/min) as compared to standard values in control carotid arteries (2.3 mL/min) [26,27], was established as a threshold for AVF maturation. A dichotomous response of the flow rate in the AVFs was observed after 7 days, with three out of four AVFs of the >23 mL/min group maintaining a higher flow rate at 14 days post-surgery. It is important to note that the current flow measurements are based on the mean of the maximum velocities in the profile and lumen diameter, which might overestimate flow values depending on the flow profile. Concerning the longitudinal development of the murine AVF post-surgery, it is known that sufficient outward remodeling and intimal hyperplasia may lead to functional maturation within the first two weeks [9]. Between 14 and 28 days after surgery, patency rates rapidly decline rather than allowing for late maturation, demonstrating the classic hallmarks of failed AVFs such as lumen loss and smooth muscle cell abundance in the intima [43]. In this context, previous therapeutic and mechanistic studies employing this murine AVF model have adhered to the gold standard follow-up of 14 days after surgery to examine the process of AVF (non-)maturation [3,44,45].

Regarding AVF velocity measurements, the relative compactness of a mouse model compared to the human situation might have affected the discriminatory potential via the variance in measurement site over time. In addition, the transition from more conventional laminar flow upstream in the afferent artery to turbulent flow at the site of anastomosis could influence the velocity indices data. This phenomenon has also been known in humans, where PSVs in turbulent sections near the fistula may result in highly variable velocities [42].

In conclusion, we used ultra-high-frequency ultrasound imaging to quantify pathophysiological remodeling processes in vein graft and AVF animal models over time. Clinically relevant parameters were adopted and novel methods established. In combination with conventional techniques, the standardized use of ultra-high-frequency ultrasound imaging represents a desirable and indispensable advancement in preclinical vascular research.

## 4. Materials and Methods

### 4.1. Animals and Surgery

This study was performed in compliance with the Dutch government guidelines and Directive 2010/63/EU of the European Parliament. All experiments were approved by the institutional committee for animal welfare of the Leiden University Medical Center (LUMC), licensed under project numbers 1160020172409 and 1160020197505. In-house bred transgenic ApoE3*Leiden mice were fed ad libitum with a diet consisting of 1% cholesterol and 0.5% cholate (Ssniff Spezialdiäten GmbH, Soest, Germany) to induce plasma hypercholesterolemia. Wildtype C57BL/6J mice were fed ad libitum with a regular chow diet.

Caval veins were obtained from donor animals from the same strain and engrafted in the carotid artery of recipient mice, both anesthetized with intraperitoneal injections consisting of a combination of midazolam (5 mg/kg, Aurobindo, Baarn, the Netherlands), medetomidine (0.5 mg/kg, Orion Corporation, Espoo, Finland), and fentanyl (0.05 mg/kg, Hameln). Toe pinching was used to validate proper anesthesia depth. The caval veins were short-term stored in heparinized water suitable for injection (100 IU/mL) prior to engraftment in the carotid artery of C57BL/6J and ApoE3*Leiden mice, as previously described [6,46]. After surgery, mice were antagonized with atipamezole (2.5 mg/kg, Orion Corporation) and flumazenil (0.5 mg/kg, Fresenius Kabi, Bad Homburg, Germany). Buprenorphine (0.1 mg/kg, MSD Animal Health, Kenilworth, NJ, USA) was used as an analgesic post-surgery and upon indication. After 28 days, the mice were anesthetized and vein grafts were harvested.

Prior to AVF surgery, C57BL/6J mice were anesthetized via intraperitoneal injections of midazolam (5 mg/kg, Aurobindo), medetomidine (0.5 mg/kg, Orion Corporation), and fentanyl (0.05 mg/kg, Hameln). After toe pinching, the end-to-side AVF ligation was performed, as previously described through the connection of the venous end of the external jugular vein to the common carotid artery [3,10]. After surgery, the mice were antagonized with atipamezole (2.5 mg/kg, Orion Corporation), flumazenil (0.5 mg/kg, Fresenius Kabi), and buprenorphine (0.1 mg/kg, MSD Animal Health). Buprenorphine was also administered on indication. After 14 days, the mice were anesthetized and AVFs were harvested.

### 4.2. Ultrasound Imaging

The animals were anesthetized with isoflurane and placed on the heated animal imaging platform of the Vevo 3100 system (Fujifilm VisualSonics, Toronto, ON, Canada), where temperature, heart rate, and respiration rate were monitored over time. During the experiments, anesthesia was maintained using a vaporized isoflurane (1 L/min of oxygen 0.3 L/min air and 2.5% isoflurane) gas system. Anesthesia depth was evaluated via toe pinching. Next, the hair in the neck region was removed with a depilatory cream. The mouse was positioned in right lateral recumbency and the transducer was aligned perpendicularly. The ultrasound acquisition of the vein grafts was performed with the MX550S transducer at an operating frequency of 25–55 MHz with an axial resolution of 40 µm. The cardiology imaging mode was used with a two-dimensional gain of 29 dB, a temporal resolution of 199–232 fps, and the following field of view: depth 10–12 mm, width 12.08–14.08 mm. The vein grafted animal groups underwent ultrasound imaging three days post-surgery, after which weekly image acquisitions took place between day 7 and day 28. For the AVF group, image acquisition took place on days 7 and 14 post-surgery. The following recordings were obtained at each time point: (i) ECG-gated Kilohertz visualization brightness-mode (B-mode); (ii) short-axis three-dimensional B-mode; (iii) long-axis three-dimensional color Doppler; (iv) caudal, intermediate, and cranial PW Doppler or inflow and anastomotic PW Doppler. For the PW Doppler measurements, an angle of 55–59° was employed.

### 4.3. Ultrasound Data Analyses

#### 4.3.1. Morphological Quantification of Vein Graft

The standard ultrasound imaging mode, B-mode, consists of cross-sectional images of the target tissue. Specifically, the ultrasound transducer transmits high-frequency sound waves through the region of interest, after which it processes reflected signals into grey-scale live images [15]. The two-dimensional area as well as three-dimensional volume measurements were derived from short-axis B-mode recordings.

The two-dimensional short-axis area quantifications were measured on three individual, vessel angulation-corrected short-axis sections at, respectively, the cranial, intermediate, and caudal level of three-dimensional B-mode vein graft recordings. Following this step, the measurements of total vein graft and lumen area were performed at all three positions and subsequently averaged for each time point (t = 3 days, t = 7 days, t = 14 days, t = 21 days, and t = 28 days). This method resulted in mean values for both lumen areas and total vein graft areas per mouse per time point, after which the former was subtracted from the latter to calculate the vessel wall area.

Regarding the three-dimensional volume measurements, the same dataset was analyzed via a different technique. The chosen three-dimensional short-axis recording, either a short-axis B-mode or its short-axis color Doppler counterpart recording, was transformed into two-dimensional short-axis slices. Then, the outer wall and lumen were manually traced in these slices every 0.04–0.12 mm, to be eventually semi-automatically merged into a three-dimensional render of both lumen and total vein graft volume. Similarly to the area analyses, the vessel wall volume was calculated by subtracting lumen volume from total vein graft volume. To ensure inter-mice and intra-mouse comparability over time, a volume per length value was used.

#### 4.3.2. Hemodynamic Quantification

PW Doppler recordings were made at the cranial, intermediate, and caudal levels of murine vein grafts. Each measurement consisted of a peak systolic velocity (PSV), mean velocity, and end-diastolic velocity (EDV) averaged from 3–5 cardiac cycle wave patterns. With these parameters, the PI and RI were calculated, using the following formulas: (i) $PI=\frac{\left(PSV-EDV\right)}{Mean\;velocity}$, (ii) $RI=\frac{PSV-EDV}{PSV}$. A representative cycle wave was chosen to serve for both maximum accelerations (ACCmax) and mean systolic acceleration (ACCsys) determinations. The flow rate was calculated in mL/min, by assuming the lumen area to be circular, according to the following equation: $flow\;rate=\left(\pi r^{2}\right)*\left(mean\;velocity\right)*\;\left(60\right)$. Spectral broadening is an ultrasound artifact characterized by a broadening of the spectral waveform, which may represent the disturbed blood flow in diseased or remodeled vessels. The spectral broadening index (SBI), normalized for peak velocity, was calculated using the following formula: $SBI=\frac{\left(spectral\;broadening\right)}{PSV}$.

### 4.4. Statistics and Software

Assessment and quantification of ultrasound recordings were performed in VEVO Lab 5.5.0 (Fujifilm VisualSonics) ultrasound analysis software. All statistical analyses were performed in Graphpad Prism 8.0. Variance in the repeated measures was tested via ANOVA. Unpaired and paired *t*-tests were run where appropriate and statistical significance was set at *p* = 0.05, graphically represented as * *p* < 0.05, ** *p* < 0.01, *** *p* < 0.001, **** *p* < 0.0001.

## Figures and Tables

**Figure 1 ijms-23-13298-f001:**
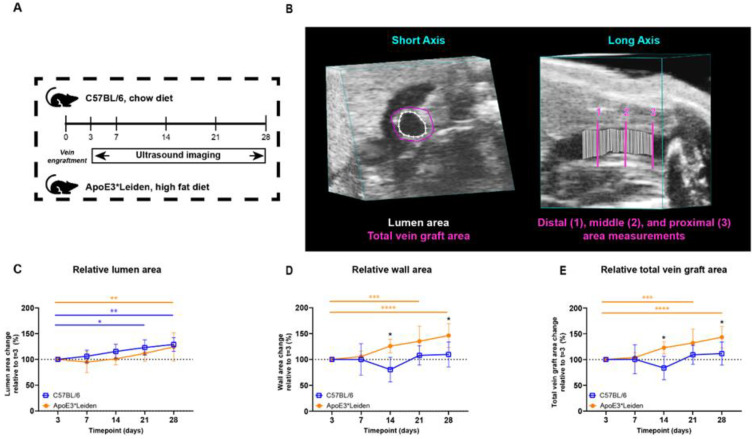
Study design and area measurements of the vein grafts in C57BL/6J and ApoE3*Leiden mice over time. (**A**) C57BL/6J (*n* = 7, chow diet) and ApoE3*Leiden mice (*n* = 7, high-fat diet) underwent vein graft surgery and were imaged on day 3 and a weekly schedule thereafter during the time course of 28 days. (**B**) Example of an area measurement, three-dimensionally corrected, depicting lumen area (white), total vein graft area (pink), and a long-axis view of the three separate measurement sites. 1 = distal, 2 = middle, 3 = proximal. Wall area is calculated by subtracting the lumen area from the total vein graft area. The total vein graft area is a measure for outward remodeling. (**C**–**E**) Graphical representations of, respectively, lumen (**C**), wall (**D**), and total vein graft area (**E**), calculated relative to their baseline measurements at t = 3 days post-surgery. Statistics: ANOVA mixed model, corrected for multiple testing; * *p* < 0.05, ** *p* < 0.01, *** *p* < 0.001, **** *p* < 0.0001.

**Figure 2 ijms-23-13298-f002:**
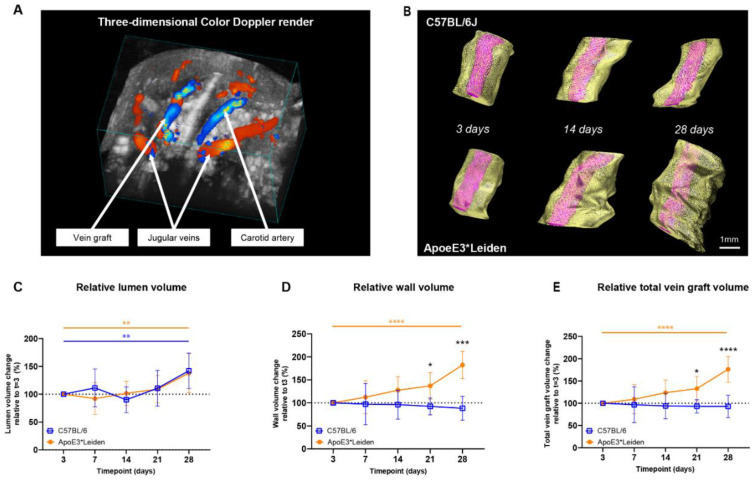
Volume measurements in the vein grafts of C57BL/6J and ApoE3*Leiden mice over time. (**A**) A three-dimensional color Doppler maximal intensity projection of the neck region of a vein grafted mouse. The nearest panel of the cube depicts the caudal side. (**B**) Representative vein graft three-dimensional renders of C57BL/6J mice and ApoE3*Leiden mice, depicting vascular remodeling over time. (**C**–**E**) Graphical representations of, respectively, lumen volume (**C**), wall volume (**D**), and total vein graft volume (**E**) of C57BL/6J (*n* = 7, chow diet) and ApoE3*Leiden mice (*n* = 7, high-fat diet), calculated relative to their baseline measurements at t = 3 days post-surgery. Total vein graft volume is a measure for outward remodeling. Volume calculations were corrected through a volume-per-length approach. Statistics: ANOVA mixed model, corrected for multiple testing; * *p* < 0.05, ** *p* < 0.01, *** *p* < 0.001, **** *p* < 0.0001.

**Figure 3 ijms-23-13298-f003:**
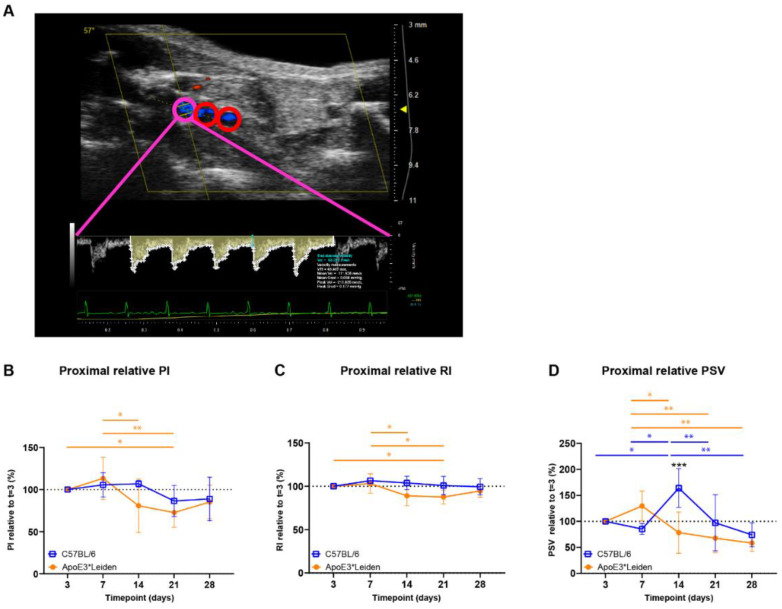
Blood flow velocity indices in vein grafts of C57BL/6J and ApoE3*Leiden mice over time. (**A**) Flow pattern of the proximal part of a murine vein graft (pink), with the middle and distal part of the vessel showing in red. The velocity–time curve of the proximal part of the vein graft is depicted in yellow, including its semi-automatically calculated velocities (white) and end-diastolic velocity (blue). (**B**–**D**) Graphical representations of the proximal PI, RI, and PSV, calculated relative to their baseline measurements at 3 days post-surgery in vein grafts of C57BL/6J and ApoE3*Leiden mice. Statistics: ANOVA mixed model, corrected for multiple testing; * *p* < 0.05, ** *p* < 0.01, *** *p* < 0.001.

**Figure 4 ijms-23-13298-f004:**
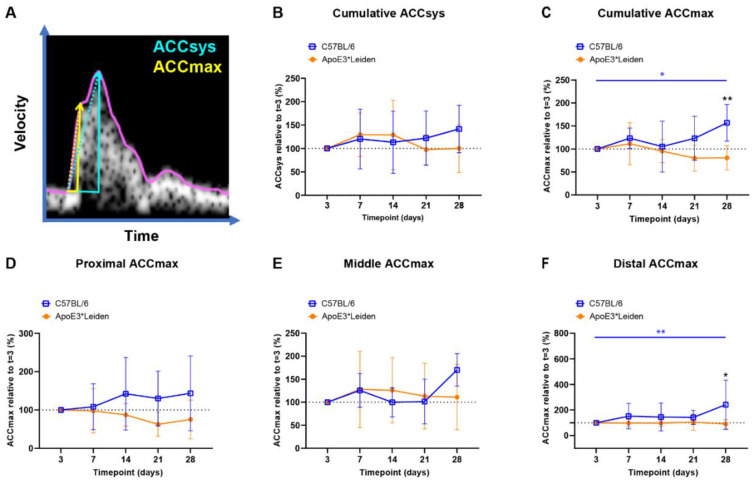
Blood flow acceleration in vein grafts in C57BL/6J and ApoE3*Leiden mice over time. (**A**) Example of a velocity–time curve of the blood flow in a murine vein graft during one cardiac cycle, demonstrating the difference in slope between ACCSys (blue) and ACCmax (yellow). (**B**,**C**) Cumulative relative ACCsys and ACCmax in vein grafts in C57BL/6J (*n* = 7) and ApoE3*Leiden mice (*n* = 7) for 28 days, relative to the baseline measurement at 3 days after surgery. (**D**–**F**) Relative ACCmax values for, respectively, the proximal (**D**), middle (**E**), and distal (**F**) part of the vein grafts of C57BL/6J and ApoE3*Leiden mice. Statistics: ANOVA mixed model, corrected for multiple testing; * *p* < 0.05, ** *p* < 0.01.

**Figure 5 ijms-23-13298-f005:**
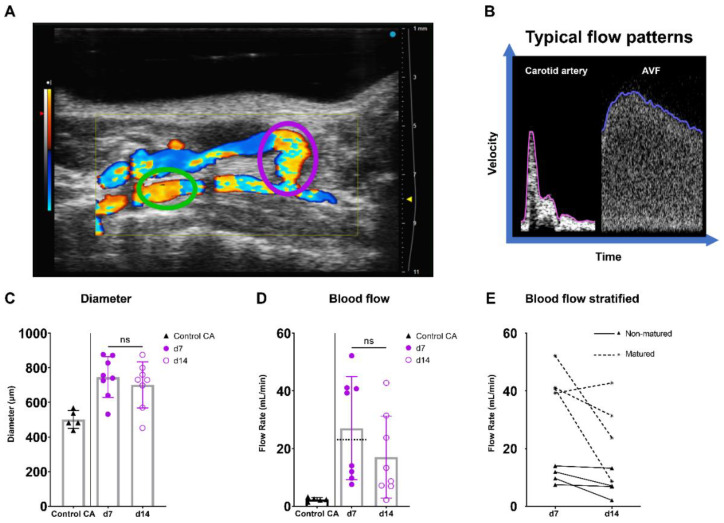
Diameter and blood flow rate measurements in AVFs in C57BL/6J mice over time. (**A**) Two-dimensional color Doppler recording of a murine AVF, with a pink oval indicating the location of anastomosis measurements and a green oval indicating the location of inflow artery measurements. (**B**) A typical velocity–time curve in a carotid artery (Control CA) and AVF in C57BL/6J mice. (**C**) Lumen diameter of control carotid arteries (*n* = 5) and murine AVFs (site of anastomosis) (*n* = 8) at 7 and 14 days after surgery. (**D**) Blood flow rate measurements of control carotid arteries (*n* = 5) and AVFs (*n* = 5) on 7 and 14 days after surgery. The dotted line indicates the cut-off value for fistula maturation (23 mL/min) (**E**) Blood flow rate measurements in the AVFs, stratified for AVF maturation or non-maturation at t = 7 days. Statistics AVF over time: paired *t*-test.

**Figure 6 ijms-23-13298-f006:**
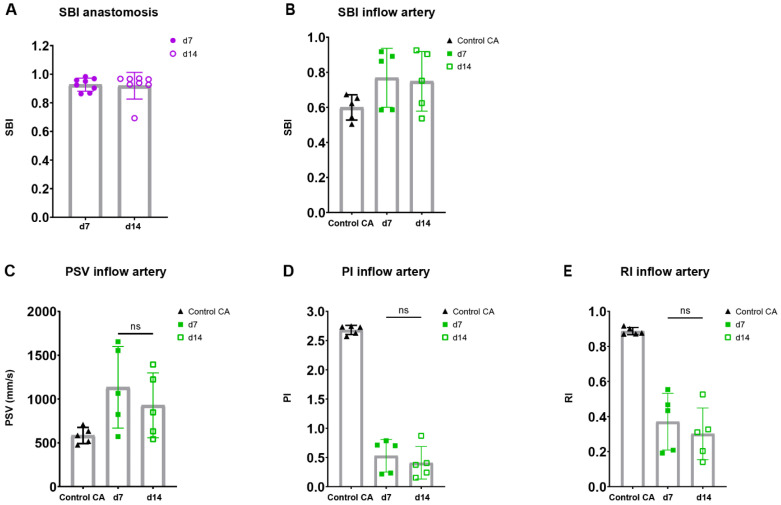
Flow pattern and velocity measurements in AVFs in C57BL/6J mice over time. SBI at the anastomosis (**A**) and inflow artery (**B**) site in AVFs of C57BL/6J mice at 7 and 14 days post-surgery. PSV (**C**), PI (**D**), RI (**E**) in control carotid arteries (*n* = 5) and AVFs (*n* = 5) in C57BL/6J mice after 7 and 14 days post-surgery. Statistics AVF over time: paired *t*-test.

## Data Availability

The original contributions presented in the study are included in the article/Appendix A, further inquiries can be directed to the corresponding author.

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
