# Peer review of "Visualization of Murine Vascular Remodeling and Blood Flow Dynamics by Ultra-High-Frequency Ultrasound Imaging"

_ijms, 2022, doi:10.3390/ijms232113298_

Round 1

Reviewer 1 Report

In this manuscript the authors demonstrate the use of high frequency ultrasound imaging as a tool to visualize and quantify the remodeling process in a vein graft in mice over time. My knowledge in the vein grafting field is limited, but my experience in ultrasound imaging in small animal models is extensive.

To the best of my knowledge the study doesn't present any ground breaking novel methods, but a nice demonstration of how non-invasive ultrasound imaging with an imaging system (Vevo 3100), which is widely available, can by used in a well-designed longitudinal study. The reasoning of using ultrasound rather than histology in this specific setting of vein grafting is well justified in the manuscript.

The manuscript is well written and methods and results are well presented. I have only minor things to be considered:

1. Throughout the manuscripts values are presented without any spacing between value and unit (e.g. line 106: 5mg/kg). This lack of spacing is unconventional and unless it is a journal requirement, I would suggest a more conventional reporting (e.g. 5 mg/kg).

2. More information on ultrasound parameters is necessary in order to reproduce the experiment: Imaging mode (general or cardiac), temporal resolution (frames per second), 2D gain, field of view, velocity threshold for PW Doppler.   

Author Response

Dear reviewer,

Thank you for your elaborate comments.
Please find attached our reply to your review report.

Best regards,

Margreet de Vries, PhD

Reviewer 2 Report

This is a thoroughly conducted  and well written study to show the practicality of measuring vein characteristics and flow in mice to study the effect of haemodynamics on veins in grafts and AVFs. Both the methodology and results are interesting.

I have only one comment.  The Doppler measurements  (para 2.3.2) describe PI as (PSV-EDV)/Mean.  In fact PI is conventionally (PSV-Min)/Mean. Reference 12 uses min but also describes RI as PSV-MinPSV when in fact RI is (PSV_EDV)/PSV as the current author describes.  It doesn’t matter in this study since EDV always appears to be the minimum btu there is confusion in the literature.  

The more significant question is over measurement of flow rate.  The mean described here is the outline trace as shown in the images, the mean of the maximum velocities in the profile.  True flow is calculated as the weighted mean of the spectrum x cross sectional area.  The current measurement overestimates flow depending on the flow profile.

I don’t know how you get around this last point but it might be worth a comment.  The measured flow rate here is an approximation.

Author Response

(The authors gave the same response as above.)
